# Preparation of Multicycle GO/TiO₂ Composite Photocatalyst and Study on Degradation of Methylene Blue Synthetic Wastewater

**Zhongtian Fu [1], Song Zhang [2] and Zhongxue Fu [3],***

[1] Key Laboratory of Ministry of Education on Safe Mining of Deep Metal Mines, Department of Environment Engineering, School of Resources and Civil Engineering, Northeastern University, Shenyang 110819, Liaoning, China

[2] School of Resources and Civil Engineering, Northeastern University, Shenyang 110819, Liaoning, China

[3] College of Mechatronics and Control Engineering, Shen Zhen University, Shen Zhen 518060, China

[*] Correspondence: fzxcn@szu.edu.cn; Tel.: +86-0755-26536224

**Abstract:** A series of composite photocatalysts were prepared by using graphene oxide (GO) prepared by modified Hummers method and TiO₂ hydrogel prepared by using butyl titanate as raw materials. The composite photocatalyst was characterized through scanning electron microscope (SEM), x ray diffraction (XRD), and Raman spectroscopy, and the degradation effect of pure TiO₂ and composite photocatalyst on methylene blue (MB) dye wastewater under different experimental conditions was studied. The results showed that TiO₂ in composite photocatalyst was mainly anatase phase and its photocatalytic activity was better than pure TiO₂. When the addition of GO reached 15 wt%, the photocatalytic activity was the highest. When 200 mg composite photocatalyst was added to 200 mL synthetic wastewater with a concentration of 10 mg/L and an initial pH of about 8, the degradation rate could reach 95.8% after 2.5 h. It is presumed that the photogenerated charges of GO/TiO₂ composite photocatalyst may directly destroy the luminescent groups in the MB molecule and thus decolorize the wastewater, and no other new luminescent groups are generated during the treatment.

**Keywords:** GO-TiO₂; photocatalytic activity; methylene blue; ultraviolet light

## 1. Introduction

In recent years, photocatalytic treatment of refractory organic pollutants has attracted increasing attention in the field of water pollution control due to its many advantages, and TiO₂ has especially been widely used owing to chemical stability, non-toxicity, easy synthesis, high reactivity, and environmental friendliness [1]. However, there are two major bottlenecks for TiO₂ as a photocatalyst: first, the utilization rate of visible light is low, TiO₂ can only absorb ultraviolet light; second, the photo-electron-hole pairs in TiO₂ particles are easy to compound, resulting in low photoquantum efficiency and poor photocatalytic activity [2,3]. At present, the methods of expanding the TiO₂ photoresponse and suppressing the carrier recombination are mainly to modify TiO₂ by various means [4,5]. In addition, TiO₂ is loaded on other functional materials to promote the separation of photo-induced charges and prolong the recombination time of photogenerated charge, which is also an effective way to improve TiO₂ photocatalytic performance and improve wastewater treatment effect [6]. Graphene (GR) is a two-dimensional structure consisting of a single layer of hexagonal lattice of carbon atoms, which are covalently bonded to each other. This structure gives excellent electrical conductivity and thermal conductivity, and high mechanical strength and large specific surface area of GR [7,8] makes GR a promising candidate for use in sensors [9,10], energy storage

and transfer [11,12], nanoelectronics [13], supercapacitors [14], composite materials [15], and other areas. Graphene oxide (GO), an oxide of GR, still retains the layered structure of GR after oxidization. However, many oxygen-containing functional groups are introduced to the GO monolith of each layer. It is generally considered that hydroxyl and epoxy groups are distributed on GO monolith, while carboxyl and carbonyl groups introduced to the edge of the monolith. The introduction of these oxygen-containing functional groups increases the complexity of single GR structure. Therefore, in addition to some properties of GR, it also has a lot of specific features, such as excellent adsorption properties [16,17]. If $TiO_2$ is combined with GO, the composite can effectively promote the separation and migration of photogenerated charges due to excellent electron conduction of GO, and can enrich organic substances within the wastewater to the surface of the composite photocatalyst due to excellent adsorption performance of GO, thus increasing the possibilities of contact with the photocatalyst and achieving better degradation and removal of these contaminants [18,19].

In this study, GO was prepared with a modified Hummers method and ultrasonic mechanical delamination, while $TiO_2$ nanoparticle was prepared with the sol-gel method. Accordingly, the $GO/TiO_2$ composite photocatalyst with different proportions of GO was prepared by the sol-gel method and was characterized by scanning electron microscope (SEM), x ray diffraction (XRD), and Raman spectroscopy. The methylene blue (MB) synthetic dye wastewater was taken as the target pollutant, in an attempt to study the effect on the treatment of synthetic dye wastewater by photocatalytic treatment process under UV irradiation, especially in terms of organic molecular degradation. Most studies in the past on treatment of MB synthetic wastewater were focused on the probability to treat this kind of water using the composite photocatalyst. In this study, the treatment process of MB synthetic wastewater by the composite photocatalyst under different reaction conditions was studied in detail. At the same time, the reaction mechanism of using the composite photocatalyst to treat MB synthetic wastewater was preliminarily discussed.

## 2. Materials and Methods

### 2.1. Chemicals and Reagents

Sulfuric acid ($H_2SO_4$, >98.0%,), Hydrochloric acid (HCl, >36.0%,), Sodium hydroxide (NaOH, >98.0%), Hydrogen peroxide ($H_2O_2$, >30.0%), Ammonia ($NH_3$, >98.0%), Acetone ($C_3H_6O$), Potassium permanganate ($KMnO_4$), Methyl alcohol ($CH_3OH$, >99.5%), Ethylalcohol ($C_2H_5OH$, >96.0%), the above reagents are all produced in Tianjin Komeo Co., Ltd., Tianjin, China. Sodium nitrate (AR), Tetrabutyl titanate (AR), Graphite Powder (AR), are produced in Sinopharm Chemical Reagent Co., Ltd., Shanghai, China.

### 2.2. Characterization of Instrumentation

Photochemical reactor (Homemade), Ultraviolet light spectrophotometer (UV3600, SHIMADZU, Kagoshima, Japan), Precision electronic balance (SARTORIUS, Göttingen, Germany), pH meter (SARTORIUS, Göttingen, Germany), vacuum drying oven (DZF-6020,SHUANGXU, Shanghai, China), High pressure reactor (Beitang chemical machinery equipment, Shenyang, China), Magnetic stirrer (Shanghai INESA Scientific Instrument CO., LTD. Shanghai, China), Ultrasonic cleaner (SB-5200, NingboXinzhi, Ningbo, China), High speed centrifuge (TG16-WS, Hunan Xiangyi, Changsha, China), Scanning Electron Microscope (ZeissUltraPlus, Jena, Germany), X-Ray Diffraction (X Pertpro, PANalytical B.V, Almelo, The Netherlands), XploRA ONE (BWS465-785S, B&WTek LLC, Ltd., Delaware, USA).

### 2.3. Synthesis of Graphene Oxide (GO)

Firstly, graphite oxide was prepared by modified Hummers method, then mixed with hydrochloric acid and centrifuged at a speed of 7000–10,000 r/min, followed by the removal of sulfate ions and neutralization with deionized water. A certain amount of centrifuged graphite oxide was mixed with a

small amount of deionized water and the mixture received ultrasonic treatment for 1 h at a frequency of 45 Hz. The resultant GO solution was vacuum dried at 30 °C to obtain GO powder.

### 2.4. Synthesis of TiO$_2$ Nanocomposite

The mixture of 10 mL butyl titanate and 18 mL anhydrous ethanol was stirred for 10 min, to obtain mixed solution A; another mixture of 18 mL anhydrous ethanol, 3 mL glacial acetic acid, and 3.3 mL deionized water was stirred for 5 min, to obtain mixed solution B (A and B were simultaneously prepared). After solution A was stirred well, solution B was slowly added by dropwise to solution A at a rate of 3 mL/min and stirred for 5 min, then 0.7 mL formamide was added to the reaction system and the solution was stirred for additional 20–30 min, until a white substance generated in a pale yellow transparent solution and its fluidity gradually deteriorated. After 1 h of stirring, the solution became alcohol gel state, namely titanium dioxide alcohol gel. The gel was sealed and placed aging for 1–2 days, then calcined in a muffle furnace at 450 °C for 4 h to obtain light yellow particles doped with a few black particles. Subsequently the particles were cooled to room temperature and crushed into powder, which was the white TiO$_2$ nanocomposite.

### 2.5. Synthesis of GO/TiO$_2$ Nanocomposite

Different weights of GO was mixed with TiO$_2$ hydrogel and electromagnetically stirred for about 10 min until the mixture became alcohol gel, then the mixture was stirred with a glass rod for 10–20 min and calcined in a muffin furnace at 450 °C for 4 h. The resulting particles were cooled to room temperature and crushed into powder, making the composite photocatalyst.

### 2.6. Photocatalysis Valuation

Methylene blue (MB) solution was used as the photocatalyst substrate in the valuation. UV–vis Spectrophotometer was used to measure the absorbance under 664 nm. The photocatalyst degradation percentage was calculated using the following equation:

$$A = (C_0 - C) \times 100\%/C_0, \tag{1}$$

where A is degradation, $C_0$ (mg/L) is the initial methyl blue (MB) concentration, $C$ (mg/L) is the substrate concentration after the photocatalyst. The initial methyl blue concentration used in the valuation was 10 mg/mL, and the ultraviolet (UV) light source was 254 nm for the irradiation of the photocatalyst.

### 2.7. Treatment of Synthetic Dye Wastewater

Photocatalytic treatment of MB dye wastewater was carried out in a home-made photocatalytic reactor. The effect of GO incorporation amount on improving the photocatalytic activity of composite photocatalyst was investigated. The composite photocatalysts prepared with different GO loads were used to treat MB dye wastewater under ultraviolet light, and the optimal composite photocatalyst was screened out. Before the photocatalytic reaction, the petri dish containing 50 mL MB synthetic dye wastewater and nano photocatalyst was placed in black box for 30 min. after the adsorption equilibrium was reached, the ultraviolet light was turned on for 150 min and stirred continuously during the light so that the nano photocatalyst could be in full contact with synthetic dye wastewater. Accordingly, the effects of the amount of composite photocatalyst, the initial pH value of synthetic wastewater, and the initial concentration of synthetic wastewater on the treatment effect were investigated, and the optimal parameters were obtained under different conditions.

### 2.8. Effect of Photocatalytic Process on Methylene Blue (MB) Dye Structure

The effect of photocatalytic process on dye molecular structure was analyzed by UV-visible full-wave scanning of MB synthetic dye wastewater before and after treatment with composite photocatalyst.

## 3. Results and Discussion

### 3.1. Scanning Electron Microscope (SEM) Analysis

Figure 1a,b shows the SEM images of TiO$_2$ nanoparticle and Figure 1c,d shows the composite photocatalyst with GO.

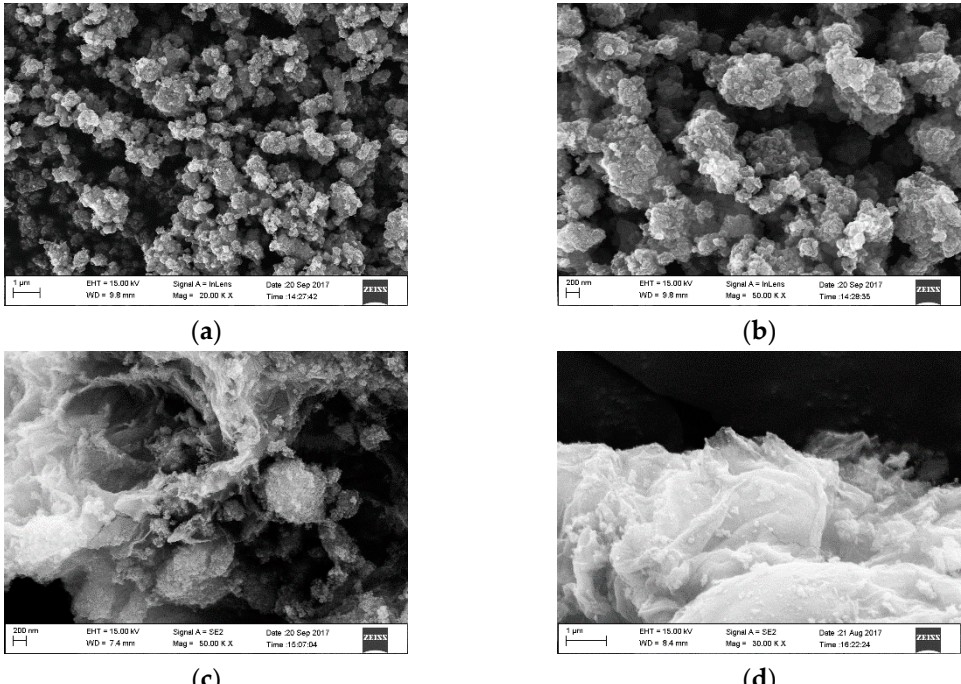

**Figure 1.** Scanning electron microscope (SEM) images of TiO$_2$ nanoparticle and composite photocatalyst with graphene oxide (GO): (**a,b**) SEM images of TiO$_2$ nanoparticle at different resolutions; (**c**) SEM images of composite photocatalyst at 3% GO content; (**d**) SEM images of composite photocatalyst at 23% GO content.

The above SEM scanning results showed that, in the absence of GO doping, TiO$_2$ powder was spherical and some particles agglomerated together to clusters. When the GO doping amount was 3%, the size of GO monolith was slightly larger than that of TiO$_2$ nanoparticles, so TiO$_2$ nanoparticles dispersed and adhered to the GO monolith. The agglomeration of spherical particles in different parts was slightly different and relatively dispersed between GO monolith structures with few connections, and the surface of composite material was mainly composed of TiO$_2$. When the GO doping amount was 23%, GO accounted for a large proportion of the formed structure, and TiO$_2$ particles adhered to GO monolith of different structures, respectively, and the formed clusters contributed to the GO monolith-like structure.

### 3.2. X Ray Diffraction (XRD) Analysis of Composite Photocatalyst

Figure 2 is an XRD pattern of TiO$_2$ and TiO$_2$-GO composite photocatalyst. As shown in Figure 2, diffraction peaks at 2θ = 25.3°, 37.8°, 48.0°, 53.9°, 62.7°, 70.3°, and 75° belonged to diffraction peaks of the (101), (004), (200), (105), (211), and (204) crystal planes of the anatase. XRD pattern revealed that the prepared TiO$_2$ is at anatase phase, with strong and sharp diffraction peak, indicating good crystallinity. The comparison of the two images demonstrated that TiO$_2$ was still present as an anatase phase in composite photocatalyst, but there was no characteristic diffraction peak of GO in the composite photocatalyst, which indicated that the presence of GO does not affect the crystal form and structure of TiO$_2$. This phenomenon may be explained by low GO content (15%) in the composite photocatalyst and covered by the diffraction peak of TiO$_2$ at 25.4° [20].

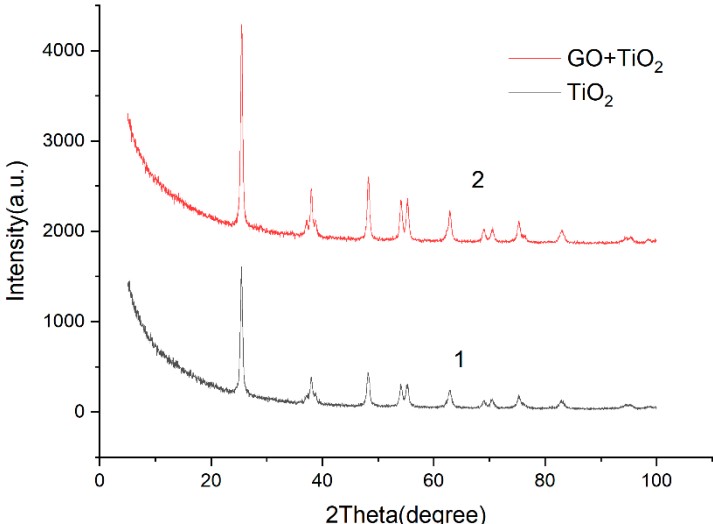

**Figure 2.** X ray diffraction (XRD) patterns of $TiO_2$ (**1**) and $TiO_2$-GO (**2**).

### 3.3. Raman Spectroscopy Analysis of Composite Photocatalyst

Figure 3 shows the Raman spectra of $TiO_2$ and $TiO_2$-GO composite photocatalysts. There were three peaks of 396 $cm^{-1}$ (B1g), 513 $cm^{-1}$ (A2g), and 639 $cm^{-1}$ (Eg) in the Raman spectrum of $TiO_2$, as the vibration mode of anatase phase $TiO_2$ [21], two peaks appeared at 1330 and 1600 $cm^{-1}$ in the Raman spectrum of $TiO_2$-GO composite photocatalysts, with $I_D/I_G$ = 1.03, which was slightly lower than that of GO ($I_D/I_G$ = 1.6) [22]. This is probably due to the GO reduction during reaction. In addition, three peaks appearing in the low frequency region were identical to those of pure $TiO_2$ vibration peak, indicating that $TiO_2$ in the composite photocatalyst is still at anatase phase.

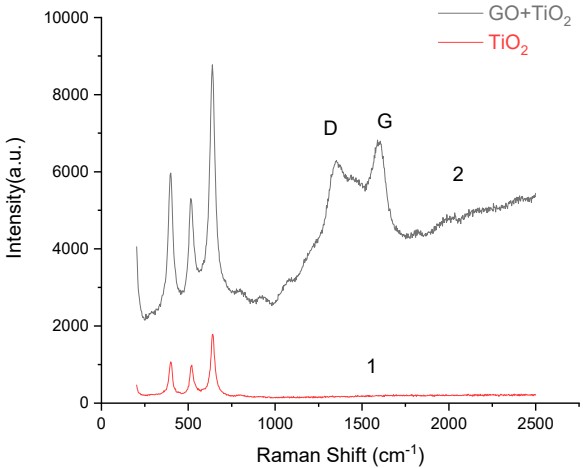

**Figure 3.** Raman spectra of $TiO_2$-GO (**1**) and $TiO_2$ (**2**).

### 3.4. Results of Photocatalytic Degradation under Different Conditions

#### 3.4.1. The Amount of GO on The Photocatalytic Activity

Figure 4 revealed the treatment effect of 50 mL MB dye wastewater at a concentration of 20 mg/L by 50 mg composite photocatalyst doped with 0, 3, 5, 8, 10, 15, 18, 20, 23, 25, and 28 wt% GO under light irradiation, absorbance value was measured every 0.5 h, a total of 2.5 h. Results showed that the photocatalytic activity when GO was added was obviously better than that of pure $TiO_2$, and the peak was observed when the amount of GO was 15 wt%. After 2.5 h of treatment, degradation rate reached 91%. When the doping amount was about 15 wt%, the GO monolith structure and $TiO_2$

nanoparticles formed a relatively uniform and reasonable structure. GO can reduce the probability of photo-generated electrons compounded with holes, and improve photocatalytic activity of $TiO_2$ due to its good conductivity. At the same time, GO can adsorb MB molecules onto the surface of the composite catalyst, increase the number of active sites, and improve the catalytic reaction effect due to its good adsorption properties. A small amount of GO doping was associated with the limitation of the catalytic reaction effect. When the amount of GO doping is relatively high, a considerable part of spherical $TiO_2$ nanoparticles entered the structure mainly formed by GO, while GO was not subject to reduction, exerted poor light transmittance, and adsorbed more dye molecules onto the surface of the catalyst, leading to scarce exposure of internal $TiO_2$ particles to ultraviolet light and a reduced probability of photogenerated charges separating from the catalyst surface. With excellent conductivity, a high GR content may promote the recombination of photogenerated electrons and holes, cause deactivation of the photocatalyst, and therefore, the treatment effect of the synthetic dye wastewater tends to decrease.

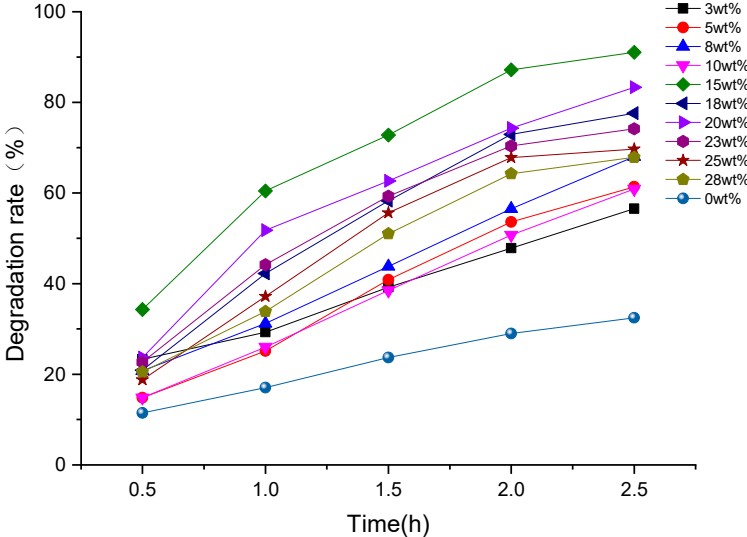

**Figure 4.** Degradation of methylene blue (MB) dye wastewater treated by photocatalyst containing different GO contents.

### 3.4.2. Effect of the Amount of Photocatalyst on the Treatment Effect

Figure 5 shows the degradation results of 200 mL MB dye wastewater at an initial concentration of 20 mg/L after 2.5 h of photocatalytic treatment of 50, 100, 150, 200, and 250 mg composite photocatalyst. The initial pH value was about 7.78. As shown in Figure 5, with the increasing amount of composite photocatalyst, the degradation rate also increased, but when the amount of composite photocatalyst ranged from 250 to 300 mg, the degradation rate began to decline, so the optimal removal effect is achieved at 250 mg photocatalyst, and the degradation rate reaches 95%. When the amount of catalyst added achieved 250 mg, the degradation began to decline, because too much catalyst caused too many suspended solids in the wastewater, and the light translucency was very low.

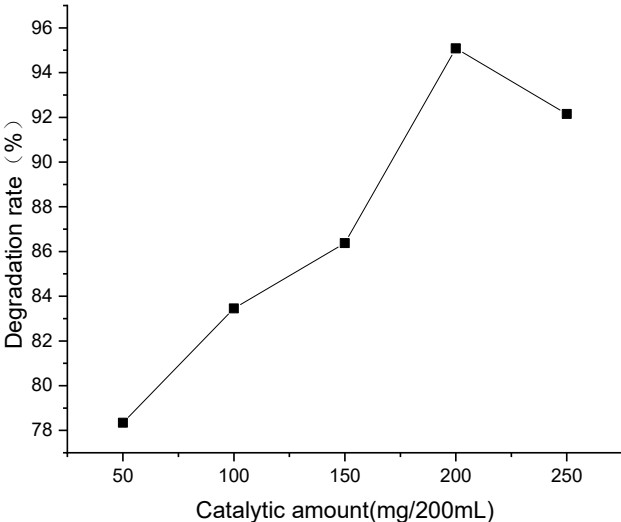

**Figure 5.** Degradation results of MB synthetic wastewater treated with different amounts of composite photocatalyst added.

### 3.4.3. Effect of Wastewater pH Value on the Photocatalytic Effect

Figure 6 shows the treatment of 250 mL MB dye wastewater at an initial concentration of 20 mg/L and different pH values (1.90, 3.98, 5.94, 7.98, and 10.02) after 2.5 h of treatment by 250 mg composite photocatalyst under ultraviolet light irradiation. The results show that the degradation rate was relatively high at pH = 5.94 and 7.98, while relatively low at pH = 1.90. When the pH value of the synthetic wastewater was 7.98, the degradation rate reached the peak, up to 94.1%. When the pH reached 10, the degradation rate began to decline significantly, which was still higher than the degradation rate at pH = 1.90. The pH value had a great influence on the stability of dye molecules, and the negative charges on the surface of GO. The GO surface was highly negatively charged in the pH range from 5.0 to 12.0, which might be ascribed to the deprotonation of anionic oxygenous functional groups on GO [23]. Highly negatively charged GO sheets in suspension could not only disperse homogeneously via electrostatic repulsion but also enhance the electrostatic attraction between them and cationic MB molecules, obviously improving the adsorption efficiency [24–27]. However, when too many dye molecules are adsorbed on the surface of GO, the active sites on the surface of the catalyst will be over occupied, which will affect the degradation process of dye molecules. At the same time, the pH value affected the formation of free radicals, and the dispersion, activity, and surface charge of photocatalysts during the photocatalytic process [28], and was related to the "isoelectric point" of Ti. According to previous data [29], the isoelectric point of GO/TiO$_2$ composite was pH = 6.7. In the solution of pH > 6.7, the surface of GO/TiO$_2$ composite was negatively charged, and the surface of catalyst was positively charged. MB molecules are organic cations, which are negatively charged in alkaline medium. In contrast, GO/TiO$_2$ composite has stronger adsorptive capacity for MB molecules. Therefore, the degradation effect is better under mild alkaline conditions.

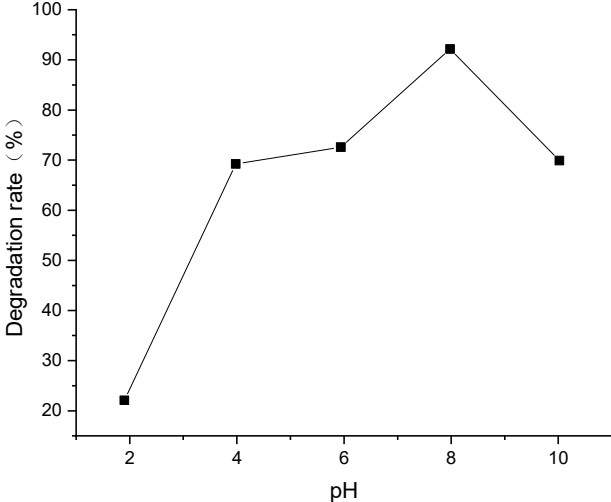

**Figure 6.** Degradation results of MB synthetic dye wastewater at different initial pH values.

3.4.4. Effect of MB Dye Wastewater Concentration on the Treatment Effect

Figure 7 shows the treatment effect of 200 mL MB dye wastewater at initial concentrations of 10, 20, 30, 40, 50, 60, 70, and 80 mg/L, after 2.5 h of 200 mg composite photocatalyst treatment. The degradation effect of MB dye wastewater peaked at the concentration of 10 mg/L. If the concentration of synthetic wastewater is too high, the transparency of wastewater will be reduced, the light will difficult to enter the wastewater and react with the photocatalyst, and the dye molecules will accumulate on the surface of the photocatalyst, which exceeds the degradation ability of the active site on the surface of the catalyst. Therefore, the degradation rate of wastewater will be reduced. It is concluded that such amount of composite photocatalyst is suitable to treat lower concentrations of synthetic wastewater.

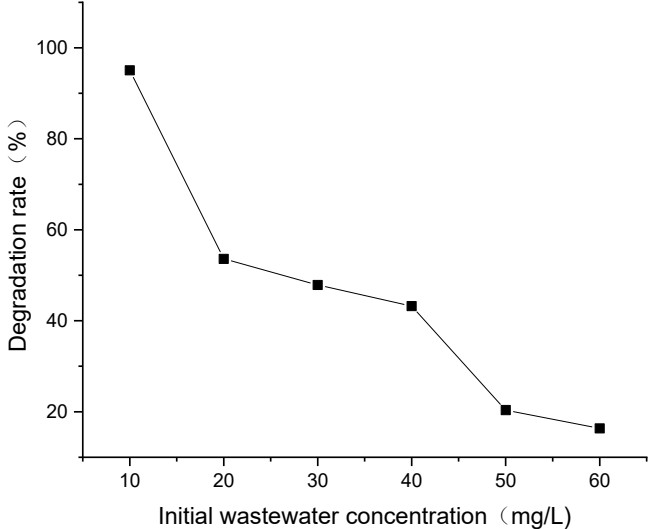

**Figure 7.** Degradation result on different concentrations of MB dye wastewater treated by composite photocatalyst under UV irradiation.

3.4.5. Repeated Experimental Results under the Optimum Conditions

From the above conditional experiments, the optimum conditions for the treatment of MB simulated wastewater with composite photocatalyst are obtained, the best GO composite ratio is 15 wt%, the most reasonable amount of composite catalyst is 200 mg/200 mL, the most suitable initial pH is better under mild alkaline conditions, and the composite photocatalyst is suitable to treat lower

concentrations of synthetic wastewater. In order to study the reproducibility of the resultant composite photocatalyst, the cycling number of the photocatalysis was record which is shown in Figure 8.

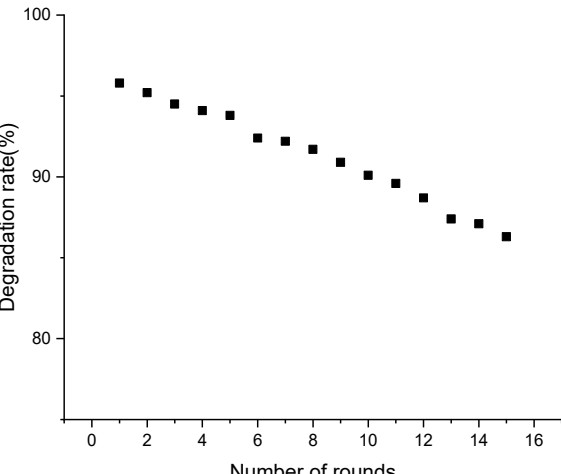

**Figure 8.** Cycling runs of the composite photocatalyst.

When each run of the photocatalysis test was fixed at 150 min for the composite photocatalyst, it can be seen from the results that composite photocatalyst can keep the degradation rates more than 85% at less for 15 cycles. There was slight decrease of the degradation rates during the cycles, which we think might due to the loss of the smaller nanoparticles which could not be collected from the centrifugation. Besides, in the process of photocatalytic wastewater treatment, it is possible to have some reactants or intermediate products remaining on the surface of the catalyst to cover the active site, resulting in a gradual decrease in photocatalytic activity. However, even after the 15th run, the degradation rate could also be more than 85%. These results indicate that the composite photocatalyst is a stable photocatalyst with good cycling life and could be used as the organic pollutants degradation

### 3.5. Photocatalytic Mechanism Analysis

The illustration of the Graphene Oxide (GO) supported $TiO_2$ as the photocatalyst is shown in Figure 9.

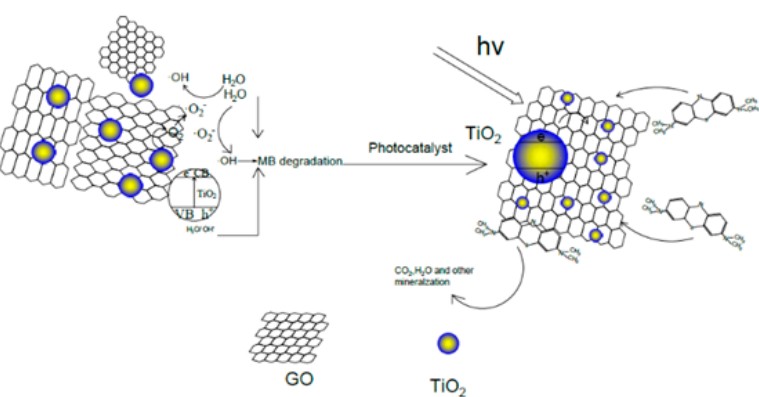

**Figure 9.** Illustration of the GO supported $TiO_2$ for the photocatalyst.

As can be seen from Figure 10, with the extension of processing time, the characteristic absorption peak intensity of MB was getting lower, indicating that the degradation effect of composite photocatalyst increased with the processing time. In addition, only the initial characteristic peak intensity of MB gradually decreased throughout the treatment process, and no new characteristic peak appeared at the other sites. This shows that when using $GO/TiO_2$ composite photocatalyst to treat the MB synthetic

dye wastewater, the photogenerated charge directly destroys the luminescent group in MB molecules, there are no other new luminescent groups produced during the process.

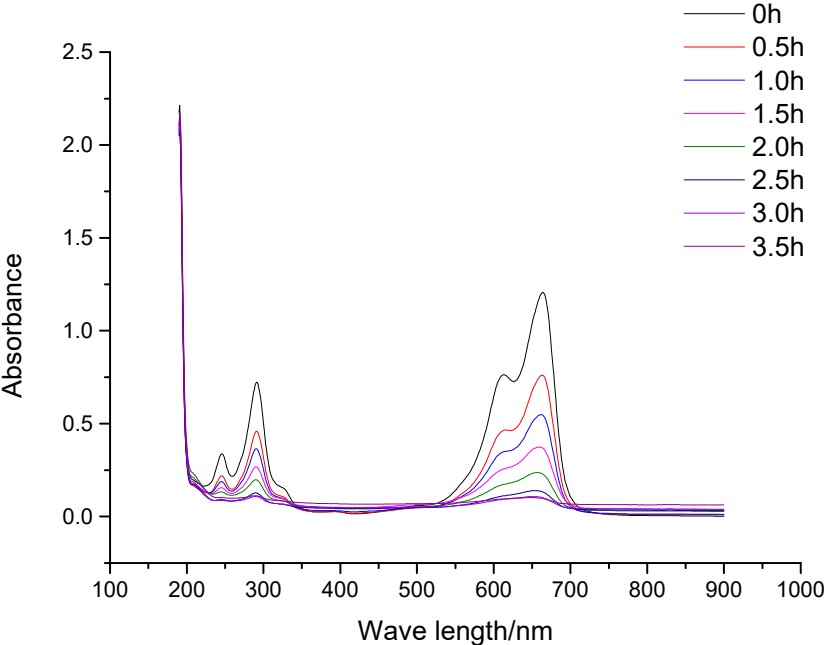

**Figure 10.** UV-visible light full-wave scanning results of MB synthetic wastewater treated by composite photocatalyst for different periods.

## 4. Conclusions

The GO-TiO$_2$ composite photocatalyst doped with different proportions of GO was prepared by the sol-gel method and was characterized by SEM, XRD, and Raman spectroscopy. MB synthetic dye wastewater was taken as the target pollutant to investigate the composite photocatalytic activity. Study results showed that TiO$_2$ in the composite photocatalyst is still mainly anatase phase, the catalytic activity is better than pure TiO$_2$, and the photocatalytic activity reaches the peak when the content of GO is 15 wt%. The composite photocatalyst is more effective for low concentrations of synthetic wastewater. When 200 mg composite photocatalyst is used to treat 200 mL synthetic wastewater at a concentration of 10 mg/L and an initial pH of about 8 for 2.5 h, the degradation rate can reach 95.8%. When the GO/TiO$_2$ composite photocatalyst treats MB wastewater, photo-generated charge directly destroys the light-emitting groups in MB molecules, there are no other new light-emitting groups produced in this process.

**Author Contributions:** Conceptualization, Z.F. (Zhongtian Fu); data curation, Z.F. (Zhongtian Fu); methodology, Z.F. (Zhongtian Fu); formal analysis, Z.F. (Zhongtian Fu); writing—original draft preparation, Z.F. (Zhongtian Fu); funding acquisition, Z.F. (Zhongxue Fu). writing—review and editing, Z.F. (Zhongtian Fu), S.Z., and Z.F. (Zhongxue Fu).

**Funding:** This work was supported by the Natural Science Foundation of Guangdong Province (2014A030310261), Shen Zhen Overseas High-level Talents Innovation and Entrepreneurship Special Fund (KQJSCX20160226174209) and Seed Funding from the Scientific and Technical Innovation Council of Shenzhen Government under Grant (JCYJ20150625102859322).

**Conflicts of Interest:** The authors declare no conflict of interest.

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
