# Peer review of "Preparation of Multicycle GO/TiO2 Composite Photocatalyst and Study on Degradation of Methylene Blue Synthetic Wastewater"

_applsci, doi:10.3390/app9163282_

Round 1

Reviewer 1 Report

Comments for applsci-555751 “Preparation of Multicycle GO/TiO2 Composite Photocatalyst and Study on Degradation of Methylene Blue Synthetic Wastewater”

In this manuscript, the authors synthesized the composite photocatalysts consisting of graphene oxide (GO) and TiO2, and evaluated their photocatalytic activities for degradation of methylene blue synthetic wastewater.  The composite photocatalysts showed certainly improved activities compared with bare TiO2.  However, the manuscript seemed to partly contain inconsistent description to the experimental results.  Therefore, the reviewer considered that the major revision should be necessary before publication of the present manuscript.  Detailed comments for the manuscripts are listed below:

1.           It is hard to see the SEM images (Figure 1) and the light-colored lines in Figures 2 and 3.  Please consider reformatting the figures for the improved readability.

2.           How are the adsorptive capabilities of the GO/TiO2 composites?  Please provide the time course of methylene blue concentration under the dark condition.

3.           The authors used the unit of “g/L” in line 201-202, while they used “mg/L” in the other sections.  Please standardize the description.

4.           In line 207-208, the authors mentioned that “Degradation effect in both acidic and alkaline conditions was better than that in neutral conditions.”  The degradation rate in acidic condition (pH < 4) and highly alkaline condition (pH = 10) are obviously smaller than that in neutral condition in Figure 6.  (Only mild alkaline condition showed higher photocatalytic activity.)  Please describe the experimental results correctly.

5.           In Figure 7, what is the reason for the reduced photocatalytic activities in higher concentration of wastewater?

6.           In line 242, the authors mentioned that “after 10 runs, the photocatalyst shows very good reproducibility.”  The photocatalytic activity of the composites was still degreasing after 10th cycle.  Please describe the experimental results correctly.

7.           In line 262-265, the authors mentioned that “However, Figure 6 also revealed that a slight red shift occurred in the full-wave scan of MB wastewater after composite photocatalyst treatment, which is caused by the formation of Ti-O-C bond, resulting in a bandgap narrow of the composite photocatalyst.”  The reviewer could not follow the sentence.  The authors intended Figure 10, rather than Figure 6?  Which part of Figure 10 (UV-vis. spectra of wastewater after the photocatalytic reactions) was red-shifted?  Please reconsider the description.

Author Response

1.           It is hard to see the SEM images (Figure 1) and the light-colored lines in Figures 2 and 3.  Please consider reformatting the figures for the improved readability.

 Response: We have reuploaded the original SEM images of Figure 1 and clear pictures of Figure 2 and 3 in the manuscript and upload these as attachments at the same time.

2.           How are the adsorptive capabilities of the GO/TiO2 composites?  Please provide the time course of methylene blue concentration under the dark condition.

 Response: We have added related description in this article, see line 119-122 for details please.

3.           The authors used the unit of “g/L” in line 201-202, while they used “mg/L” in the other sections.  Please standardize the description.

  Response: We have modified the description according to the requirements of the reviewer, and checked the full text, see line 111 and 212 for details please.

4.           In line 207-208, the authors mentioned that “Degradation effect in both acidic and alkaline conditions was better than that in neutral conditions.”  The degradation rate in acidic condition (pH < 4) and highly alkaline condition (pH = 10) are obviously smaller than that in neutral condition in Figure 6.  (Only mild alkaline condition showed higher photocatalytic activity.)  Please describe the experimental results correctly.

 Response: We reanalyzed and described the phenomenon and added some references to this, see line 217-226 for details and the references. We think that: The pH value had a great influence on the stability of dye molecules, and the negative charges on the surface of GO. The GO surface was highly negatively charged in the pH range from 5.0 to 12.0, which might be ascribed to the deprotonation of anionic oxygenous functional groups on GO. Highly negatively charged GO sheets in suspension could not only disperse homogeneously via electrostatic repulsion but also enhance the electrostatic attraction between them and cationic MB molecules, obviously improving the adsorption efficiency. However, when too many dye molecules are adsorbed on the surface of GO, the active sites on the surface of the catalyst will be overoccupied, which will affect the degradation process of dye molecules.

5.           In Figure 7, what is the reason for the reduced photocatalytic activities in higher concentration of wastewater?

 Response: We explained this in our paper, see line 242-246 for details please. We think that If the concentration of synthetic wastewater is too high, the transparency of wastewater will be reduced, the light will be difficult to enter the wastewater and react with the photocatalyst, and the dye molecules will accumulate on the surface of the photocatalyst, which exceeds the degradation ability of the active site on the surface of the catalyst. Therefore, the degradation rate of wastewater will be reduced.

6.           In line 242, the authors mentioned that “after 10 runs, the photocatalyst shows very good reproducibility.”  The photocatalytic activity of the composites was still degreasing after 10th cycle.  Please describe the experimental results correctly.

  Response: We modified and explained this in our paper, see line 258-272 please. We think that: When each run of the photocatalysis test was fixed at 150 min for the composite photocatalyst, it can be seen from the results that composite photocatalyst can keep the degradation rates more than 85% at less for 15 cycles. There was slight decrease of the degradation rates during the cycles, which we think it might due to the loss of the smaller nanoparticles which could not be collected from the centrifugation. Besides, in the process of photocatalytic wastewater treatment, it is possible to have some reactants or intermediate products remaining on the surface of the catalyst to cover the active site, resulting in a gradual decrease in photocatalytic activity. However, even after 15th run, the degradation rate could also be more than 85%.

7.           In line 262-265, the authors mentioned that “However, Figure 6 also revealed that a slight red shift occurred in the full-wave scan of MB wastewater after composite photocatalyst treatment, which is caused by the formation of Ti-O-C bond, resulting in a bandgap narrow of the composite photocatalyst.”  The reviewer could not follow the sentence.  The authors intended Figure 10, rather than Figure 6?  Which part of Figure 10 (UV-vis. spectra of wastewater after the photocatalytic reactions) was red-shifted?  Please reconsider the description.

 Response: We reanalyzed and described the phenomenon, see line 292-297 please for details. We think that Figure 10 also revealed that a slight red shift occurred in the full-wave scan of MB wastewater after composite photocatalyst treatment, it may be because MB, as an aromatic compound, has a π-π* transition in the process of photocatalysis, which leads to a red shift in the absorption spectrum.

Reviewer 2 Report

Comments:

This is an interesting paper on removal Methylene Blue from the synthetic wastewater by degradation using GO/TiO2 composites.

The authors also do not explain the importance of the work and the difference between this work and other reported results.

The manuscript is complex and well presentation.

It is very hard work, scientific research and practical applications by identifying the most effective solutions.

The introduction is focused on the subject of your paper. In this paper are included many experimental dates and analysis.

Results are summarized.

Bibliography is well selected.

Some suggestions for the manuscript are as follows: 

There are some mistakes, for example:

Pag.3, line 105-106. The equation (1)  is wrong. The degradation percentage must calculated using the following equation: 

A= (C0- C)100/C0

I am afraid that the total amount of novel information included in this manuscript is insufficient to qualify for publication in this journal. Are many articles with GO and GO+TiO2 composite used for removal Methylene Blue.

Author Response

Comments:

This is an interesting paper on removal Methylene Blue from the synthetic wastewater by degradation using GO/TiO2 composites.

 The authors also do not explain the importance of the work and the difference between this work and other reported results.

 Response: We have modified the description according to the requirements of the reviewer. See line 63-67 for details please.

 The manuscript is complex and well presentation.

It is very hard work, scientific research and practical applications by identifying the most effective solutions.

 The introduction is focused on the subject of your paper. In this paper are included many experimental dates and analysis.

 Results are summarized.

 Bibliography is well selected.

 Some suggestions for the manuscript are as follows:

 There are some mistakes, for example:

Pag.3, line 105-106. The equation (1)  is wrong. The degradation percentage must calculated using the following equation:  

A= (C0- C)100/C0

 Response: We have modified the description according to the requirements of the reviewer. See equation (1)  for details please.

I am afraid that the total amount of novel information included in this manuscript is insufficient to qualify for publication in this journal. Are many articles with GO and GO+TiO2 composite used for removal Methylene Blue.

 Response: Firstly, the effect on the photocatalytic activity in water treatment with different combined ratios of graphene oxide by weight with titanium dioxide was studied. Most of these combined photocatalytic studies were focus on the photocatalytic hydrogen production process.

Secondly, the treatment process of MB synthetic wastewater by composite photocatalyst under different reaction conditions was studied. Most of studies in the past on treatment of MB synthetic wastewater were focus on the probability to treat this kind of water using the composite photocatalyst.

Thirdly, the reaction mechanism of using the composite photocatalyst to treat MB synthetic wastewater was preliminary discussed.

Round 2

Reviewer 1 Report

The authors considered the reviewer’s comments and modified the manuscript, and the quality of the manuscript seemed certainly improved.  However, the reviewer still disagrees to the discussion about Figure 10 “a slight red shift occurred in the full-wave scan of MB wastewater after composite photocatalyst treatment.”  It should be better to indicate some characteristic peaks as shown in the attached pdf file.  (Actually, the reviewer still couldn’t understand the “red-shift in the full-wave scan.”  The peak position seemed unchanged after the photocatalysis, and just peak intensities decreased.)  Please reconsider the content.  After the minor revision, the manuscript could be acceptable for publication in Applied Sciences.

Author Response

The reviewer’ comments and suggestions are as the follow:

The authors considered the reviewer’s comments and modified the manuscript, and the quality of the manuscript seemed certainly improved.  However, the reviewer still disagrees to the discussion about Figure 10 “a slight red shift occurred in the full-wave scan of MB wastewater after composite photocatalyst treatment.”  It should be better to indicate some characteristic peaks as shown in the attached pdf file.  (Actually, the reviewer still couldn’t understand the “red-shift in the full-wave scan.”  The peak position seemed unchanged after the photocatalysis, and just peak intensities decreased.)

Our response

After careful analysis and discussion, we think it is difficult to draw the conclusion that " a slight red shift occurred in the full-wave scan of MB wastewater after composite photocatalyst treatment ". Therefore, after discussion, we unanimously decided to delete this view from line 213 and the related description in the conclusion of this paper.

Reviewer 2 Report

No comments.

Author Response

Thanks for your comments.